# Adaptation of the Aphasia Impact Questionnaire-21 into Greek: A Reliability and Validity Study

**Marina Charalambous** [1,2,*], **Phivos Phylactou** [2], **Alexia Kountouri** [3], **Marios Serafeim** [2], **Loukia Psychogios** [4], **Jean-Marie Annoni** [1] and **Maria Kambanaros** [2]

1 Neurology Unit, Laboratory of Cognitive and Neurological Sciences, Faculty of Science and Medicine, University of Fribourg, Chemin du Musée 8, 1700 Fribourg, Switzerland
2 The Brain and Neurorehabilitation Lab, Department of Rehabilitation Sciences, Cyprus University of Technology, Limassol 3036, Cyprus
3 "Solidarity Network Nicosia In Action" (NicInAct), Multifunctional Foundation, Nicosia Municipality, Eptanisou 11, Nicosia 1016, Cyprus
4 Euroclinic Group, Theseus Physical Medicine and Rehabilitation Center, 176 71 Athens, Greece
* Correspondence: marina.charalambous@unifr.ch

**Abstract:** The impact of aphasia on the everyday life of Greek-speaking people with aphasia (PWA) is often underestimated by rehabilitation clinicians. This study explores the adaptation and psychometric properties of the Greek (GR) version of The Aphasia Impact Questionnaire-21 (AIQ-21-GR) to address this issue. The aim of this study is to determine the reliability and validity of the Greek version of the AIQ-21. The AIQ-21-GR was administered to 69 stroke survivors, 47 with aphasia and 22 without aphasia. The data were analyzed to determine reliability and validity. Content validity was based on the Consensus-based Standards for the selection of health Measurement Instruments guidelines. The AIQ-21-GR shows high levels of reliability and validity. The results confirmed high scores of internal consistency (Cronbach's $\alpha = 0.91$) and indicated good known—groups validity (Mann–Whitney $U = 202$, $p < 001$). Content validity achieved high scores with an overall median score of 4 [$Q_{25} = 4$, $Q_{75} = 5$]. The psychometric properties of the AIQ-21-GR support the reliability and validity of the tool for investigating the impact of aphasia on the quality of life of Greek-speaking PWA. The AIQ-21-GR can be used for setting functional goals in collaboration with PWA and as a patient reported outcome measure for functional communication training.

**Keywords:** AIQ-21-GR; aphasia assessment; people with aphasia; tool validation; stroke

## 1. Introduction

In many countries, people with stroke are discharged from a medical rehabilitation setting back into society [1–3] with minimal community integration [4]. New stroke survivors face stroke-related disabilities and a broad range of psychosocial challenges that negatively impact their quality of life (QoL) [5,6]. A main area affecting QoL is the communication difficulties experienced by stroke-induced aphasia [7]. Aphasia is an acquired communication impairment that impacts the ability of the person to speak, understand, read, write, calculate, and carry out successful everyday interactions [3]. Aphasia affects approximately up to a third of all chronic stroke survivors [8,9].

### 1.1. Aphasia and QoL

Aphasia is linked to poor functional communication outcomes [7,10], little opportunity of return to work [11,12] and reduced activities of daily living (ADL) [6,13]. A high incidence of depression is also reported, with estimates ranging between 62 and 70% and higher, for PWA compared to stroke survivors without aphasia [14]. Aphasia also leads to social isolation [15] and limited participation in social and cultural events and family activities [5,7,16]. PWA also have reduced access to health and social services because

of physical and communication barriers. Therefore, it is crucial for PWA to gain back their communication skills to express their thoughts, everyday needs, emotions and enjoy meaningful interactions [10].

### 1.2. Aphasia Assessments and the ICF [17]

In Greece and Cyprus, speech-language therapists (SLTs) primarily assess the language difficulties of PWA using formal assessments that tap into the impairment (i.e., body structures and functions) domain of the International Classification on Functioning, Disability and Health Framework (ICF) [7,17]. Such assessments are primarily administered in the acute stroke and subacute phase and focus on the linguistic impairment of the person rather than his/her functional communication abilities in everyday life [7]. Upon discharge from the hospital, there is a discontinuation in the evaluation of the impact of aphasia on everyday life [7]. In the chronic phase, information on the meaningful involvement of PWA in life events and shared interactions, and the overall rating of QoL is absent as clinicians do not have tools to measure the activity and participation level of the ICF [17].

### 1.3. Patient and Public Involvement and the Development of QoL Self-Rating Tools

According to the Patient and Public Involvement (PPI) approach, to meet the practical needs of PWA, self-rating tools examining the impact of aphasia on QoL should be developed with the active engagement of PWA as members of the research team [18]. Patient Reported Outcome Measures (PROMs) related to QoL should be developed using PPI co-production methodology [3]. This is vital for the democratic representation of PWA within research teams and to develop pragmatic QoL tools [19]. In a recent scoping review by Charalambous and colleagues [3] on PPI involvement of PWA in the creation of QoL and aphasia impact-related tools, the results showed that only four out of the twenty published studies actively involved PWA as co-researchers during their development. These studies involved the development of the Assessment for Living with Aphasia [20], the Aphasia Communication Outcome Measure [21], the Measurement of Stroke Environment [22] and the Aphasia Impact Questionnaire-21 [23].

### 1.4. Aphasia Impact Questionnaire-21 (AIQ-21)

The AIQ-21 is a PROM that was created with PPI and co-production methods in the United Kingdom by Swinburn and colleagues [23]. The AIQ-21 is a self-reported questionnaire that assesses the impact of aphasia on the QoL of PWA. It followed the Social Model of Disability framework by Byng and Duchan [24] and was based on the Communication Disability Profile [25]. The AIQ-21 is a part of the revised Comprehensive Aphasia Test [26]. During its development, PWA were actively involved in the selection of the items, and they advised on format, content, and scoring. The AIQ-21 is validated in English [23] and in Turkish [27]. It has been translated into Danish, Catalan, Japanese, Dutch and Slovakian (https://www.aiq-21.net, accessed on 1 September 2022).

For the validation of the AIQ-21 in Greek we compared our scores with an equivalent gold standard tool for measuring QoL, the Stroke and Aphasia Quality of Life Scale-39 (SAQOL-39) [28]. The SAQOL-39 is adapted and psychometrically validated in Standard Greek [29,30] and measures QoL after stroke and aphasia from the patients' perspective. The Turkish version of the SAQOL-39 (SAQOL-39-TR) [31] was also used in the validation study of the Turkish version of the AIQ-21 (AIQ-21-TR) [27].

The long-term effects of aphasia on the QoL of PWA in the chronic phase should become a key priority during the rehabilitation process in Greek speaking countries. There is a gap in the evaluation of the language abilities of PWA in the acute phase and the exploration of the impact of aphasia on the QoL of Greek-speaking PWA in the chronic phase. Apart from the Greek version of the SAQOL-39, there is no other published PROM tool in Greek used to measure the impact of aphasia on QoL and voice the direct needs of PWA. This study aimed to evaluate the reliability and validity of the Greek version of the AIQ-21 (AIQ-21-GR).

## 2. Materials and Methods

For the translation and adaptation of the AIQ-21-GR we followed PPI methodology. PPI methodology was based on the "The Dialogue Model" [32]. The Dialogue Model is a multi-phased participatory approach. Author M.C. served as the facilitator. She stimulated conditions for dialogue between stakeholders during all steps of the process from linguist adaptation of the tool (e.g., from the forward translation step) to the cognitive interviews for content validity. The research team included a PPI partner, co-author A.K., a young female stroke survivor with chronic mild-moderate aphasia. She holds a master's degree in Research Methodologies but dropped out of her doctoral studies in Social Care at the University of Sussex after her stroke event. A.K. has previous experience with aphasia research as she was involved in research with Charalambous and colleagues between 2019 and 2022 [18]. The aim for collaborating with A.K. as the PPI partner, was to enhance the quality of the content, and the linguistic accurateness of the adapted version of the AIQ-21-GR. A.K. was involved in all stages of the adaptation and content validation of the tool by overseeing and reviewing the translations of the materials, validating the content of the items, and finalizing all aspects of the Greek version of the AIQ-21.

### 2.1. Participation Criteria

Participants were eligible to participate in the study if they met four pre-established inclusion criteria: (1) native Greek speakers, (2) age $\geq$ 18 years, (3) in the chronic phase, i.e., experienced a stroke at least 6 months prior to the study, (4) presented with chronic aphasia as diagnosed by an SLT. Participants were excluded if they presented with an additional diagnosis of dementia or any other degenerative disease, psychiatric comorbidity, profound hearing impairment and/or visual difficulties that would interfere with their taking part in the study or unilateral spatial neglect as detected with the Albert's Test [33]. Hearing, vision, and the medical history was determined by observation, self-report and/or reports from the care provider during the case history interview.

### 2.2. Recruitment

Participants were recruited from both Greece and Cyprus from August 2021 to June 2022. Recruitment sources were the Melathron Agoniston EOKA Neurorehabilitation Center in Limassol, the Limassol General Hospital Stroke Registry, private rehabilitation and neurology clinics in Nicosia, Limassol and Athens, private speech-language therapy clinics/offices in Cyprus and Greece, the Cyprus Stroke Association registry, and the Aphasia Communication Team groups run by the Cyprus Stroke Association. A total of 72 participants were referred to this study, with 3 later dropping out for personal reasons.

### 2.3. Sample Size

This study included 69 participants with chronic stroke. The sample size is sufficient for the scope of the research compared to previous sample sizes for validation of the AIQ-21 based on population and incidence of stroke (see Table 1). Data in Table 1 are from the Burden of Stroke report in Europe for UK, Greece, and Cyprus [34] and from the study of Ozturk et al. [35] and Köseoğlu et al. [36] for Turkey.

**Table 1.** Sample size of the AIQ-21-GR based on former versions.

| Country | Population | Incidence Estimate | AIQ-21 Sample Size (*n*) |
|---|---|---|---|
| United Kingdom | 65,542,579 | 106,000 strokes/year | *n* = 90 PWA |
| Turkey | 85,561,976 | 400,000 strokes/year | *n* = 104 (43 PWA + 61 healthy) |
| Greece & Cyprus | 11,606,813 803,147 | 28,000 strokes/year 1000 strokes/year | *n* = 69 (47 PWA + 22 stroke no aphasia) |

### 2.4. Participants

The reliability and validity of the adapted Greek version of the AIQ-21 was based on the results of 69 stroke survivors, 47 of whom were PWA and 22 individuals of whom were stroke survivors without aphasia. Participants were native Greek speakers with no visual or hearing problems that could interfere with the study's protocol. The demographic details of the participants are presented in Table 2.

**Table 2.** Demographic data for participants with and without aphasia.

| Characteristics | People with Aphasia (*n* = 47) | Stroke Survivors without Aphasia (*n* = 22) |
|---|---|---|
| *Gender* | | |
|     Male | 19 (40%) | 11 (50%) |
|     Female | 28 (60%) | 11 (50%) |
| *Age* | | |
|     Mean (sd) | 58.3 (17.5) | 55.7 (18.7) |
|     Minimum–Maximum | 22–85 | 20–73 |
| *Stroke Type* | | |
|     Ischemic | 33 (70%) | 11 (50%) |
|     Hemorrhagic | 14 (30%) | 11 (50%) |
| *Lesion Location* | | |
|     Left | 40 (85%) | 5 (23%) |
|     Right | 7 (15%) | 17 (77%) |
| *Hemiplegia* | | |
|     Left | 5 (10%) | 16 (70%) |
|     Right | 37 (80%) | 5 (25%) |
|     None | 5 (10%) | 1 (5%) |
| *Months Post Stroke Diagnosis* | | |
|     Mean (sd) | 19. 5 (20.8) | 22.2 (19.7) |
|     Range | 6–96 | 6–72 |
| *Completed Education* | | |
|     Primary | 3 (7.5%) | 4 (17.5%) |
|     Secondary | 26 (55%) | 9 (40%) |
|     College | 1 (2.5%) | 3 (12.5%) |
|     Bachelor's | 15 (30%) | 5 (25%) |
|     Master's | 2 (5%) | 1 (5%) |
| *Marital Status* | | |
|     Married | 30 (60%) | 16 (70%) |
|     Single | 10 (31%) | 4 (20%) |
|     Divorced | 7 (9%) | 2 (10%) |
| *Socioeconomic Status Based on Former Occupation* | | |
|     Higher managerial | 16 (32%) | 5 (24%) |
|     Intermediate occupation | 11 (24%) | 4 (18%) |
|     Manual occupation | 9 (20%) | 4 (18%) |
|     Unemployed | 11 (24%) | 9 (40%) |

### 2.5. Data Collection

Administration of the AIQ-21-GR and all other assessments took place either at the participant's home, or private clinics/offices. Testing was carried out by four certified Greek-speaking SLTs experienced in aphasia rehabilitation. Prior to the commencement of the study the research SLTs received training by the first author, M.C., on the administration of the AIQ-21-GR in Greek. Informed consent was obtained from each participant at the beginning of the study. Specifically, for PWA simplified information about the project was given in an aphasia friendly format with an additional aphasia friendly consent form as recommended by the AIQ-21.

### 2.6. Measures

The *Aphasia Impact Questionnaire-21*: The AIQ-21 [23] is a self-rating questionnaire that includes 21 simple questions. Each question is accompanied by black line-drawing of pictures (from 1 up to 4 pictures on each page depending on the content of the question) on a white background, to support comprehension. The AIQ-21 consists of 21 items: 6 items in the Communication domain, 7 items in the Participation domain, and 11 items in the Emotional state/Well-being domain. Total scores range from 0 to 84, with higher scores indicating higher impact of aphasia (0 = no problem to 4 = impossible). PWA score each item using a 5-point scale (0 to 4) which is indicated at the bottom of each page.

The *Aphasia Severity Rating Scale*: For the PWA group, the Aphasia Severity Rating Scale (ASRS) of the Greek adaptation of the Boston Diagnostic Aphasia Examination Short Form (BDAE-SF) [37] was used to rate the severity of the observed language difficulties. Spontaneous speech samples were elicited during a 15 min semi-structured interview that comprised four topics: the stroke story, occupation, family and housing and hobbies [38]. Aphasia severity was assessed by each SLT using the ASRS to allow a classification based on verbal output. Scores on the ASRS range from 0 to 5, with 0 revealing very severe non-fluent aphasia and 5 indicating very mild aphasic difficulties.

The *Stroke and Aphasia Quality of Life Scale-39*: The SAQOL-39 [28] measures the QoL of people with stroke and aphasia from the person's perspective. It consists of 39 items over three domains: Physical (16 items), Communication (7 items) and Psychosocial (16 items). Each item has a 5-point scale (1 to 5), and mean scores range from 1 to 5, with higher scores indicating better QoL. For PWA and stroke survivors without aphasia, we compared the AIQ-21 total scores with the overall SAQOL-39 scores and between the two corresponding domains of the SAQOL-39: Communication and Psychosocial.

### 2.7. The Adaptation Processes

This study examines the adaptation and validation of the AIQ-21-GR based on the original UK version [23]. The senior author M.K. received permission from Dr. Swinburn via the Collaboration of Aphasia Trialists consortium to translate, adapt and validate the AIQ-21 in Greek, for use in both Cyprus and Greece, in conjunction with the Greek adaptation of the Comprehensive Aphasia Test (work in progress). Adapting and validating the AIQ-21-GR involved translating and adapting the test manual from English into Greek, including instructions on administration, the scripts, the scoring sheet, and the consent forms. Standard Modern Greek (Greek) is the native language of Greeks living in Greece and is acquired at school in Cyprus, as it is the variety used in formal oral and written communication [39]. Cypriot Greek, a dialect of Standard Modern Greek, is the mother tongue of Greek Cypriots and is used in informal interactions [39]. For this reason, all materials were translated and adapted into Standard Modern Greek to serve clinicians in both Greek-speaking countries.

### 2.8. Linguistic Adaptation

To achieve item and semantic equivalence of the Greek translation of the AIQ-21 to the original UK version, the Schmidt and Bullinger [40] method was followed for linguistic adaptation of cross-cultural QoL instruments. Schmidt and Bullinger [40] identify several issues that need to be considered when conveying the features of an item created for a particular culture to a completely different culture. For this reason, several translations and steps were performed in this work, to ensure language and content validity [40]. This methodology was also applied in the Turkish study for the adaptation of the AIQ [27]. Furthermore, we performed a content validity exercise using the COnsensus-based Standards for the selection of health Measurement INstruments (COSMIN) guidelines [41].

Step 1: Forward translations of the tool and related materials into Greek

Forward translations of all materials from English into Greek were initiated. We collected translations of the AIQ-21 in Greek from (1) an English language professional interpreter from the Language Centre of the Cyprus University of Technology, (2) a bilin-

gual Greek–English linguist, (3) a bilingual Greek–English lay person, and (4) a bilingual Greek–English person with chronic aphasia. All versions were collected by M.C. and in collaboration with the PPI partner A.K. and an SLT academic from the Department of Rehabilitation Sciences of the Cyprus University of Technology (both blinded to the study) performed comparisons of the four translations. This step resulted in the first draft of the AIQ-21-GR questionnaire.

Step 2: Back translation of the tool and related materials into English

The first draft of the AIQ-21-GR was back translated into English by a bilingual Greek–English psychologist who was not familiar with the original version of the AIQ-21. M.C. and the SLT academic compared the items of the translated text with the original version and noted excellent correspondence to the original source (90% of equivalent words). This was performed to assure the accurateness and fidelity of the translation to the original.

Step 3: Expert opinion on the appropriateness of the translation

We asked two professional SLTs working in aphasia rehabilitation, one from Athens and one from Cyprus, to review the Greek version of the AIQ-21. They agreed on the proposed version and confirmed the relevance of the items included in the Greek version.

Step 4: Pilot study for language clarity in a healthy control population

For this step we initiated a short pilot study with a group of 16 bilingual Greek–English final-year SLT students who scored both the Greek and the English version of the AIQ, to ensure for language relevance of the translated Greek version. For outcomes, please see the Section 3.2.1 in the Section 3.

Step 5: Content validity analysis

As a final step we performed a content validity analysis. For this exercise, we followed the COSMIN guidelines (www.cosmin.nl, accessed on 10 February 2022). Accordingly, in tool adaptation studies where a translation of a PROM is performed, it is necessary for the research team to complete a content validity evaluation of the translated PROM [41]. For this reason, three aspects of content validity were distinguished: (1) relevance (all items in the AIQ-21-GR should be relevant for the construct of interest for PWA and the context of use), (2) comprehensiveness (no key aspects of the construct should be missing), and (3) comprehensibility (the items should be understood by PWA as intended).

Two rounds of cognitive interviewing were carried out to explore how respondents understood and answered the items with the aim of improving the validity and acceptability of the AIQ-21 questionnaire in Greek. Cognitive interviews were completed with 6 PWA, 4 stroke survivors without aphasia, 10 healthy individuals and 10 SLTs. During the first round of interviews participants were prompted to complete the AIQ-21-GR questionnaire unaided, followed by a semi-structured interview [42]. This process allowed the interviewer (author M.C.) to query the participant's understanding of an item and their interpretation of the instructions and response options. Based on comments from the respondents' minor revisions were made to the questionnaire to facilitate patient understanding of the items, e.g., in the Greek translation item 14 /ka'θolu avo'iθiτos/ had two negative words which created semantic confusion. Furthermore, for the response options on the AIQ-21-GR, PWA requested all numbers on the Likert Scale (0, 1, 2, 3, 4) appear on the bottom of the page for each question. In the original English version, participants viewed only two numbers: the number 4 (impossible) and 0 (no problem), without any of the intermediate numbers (see Figure 1).

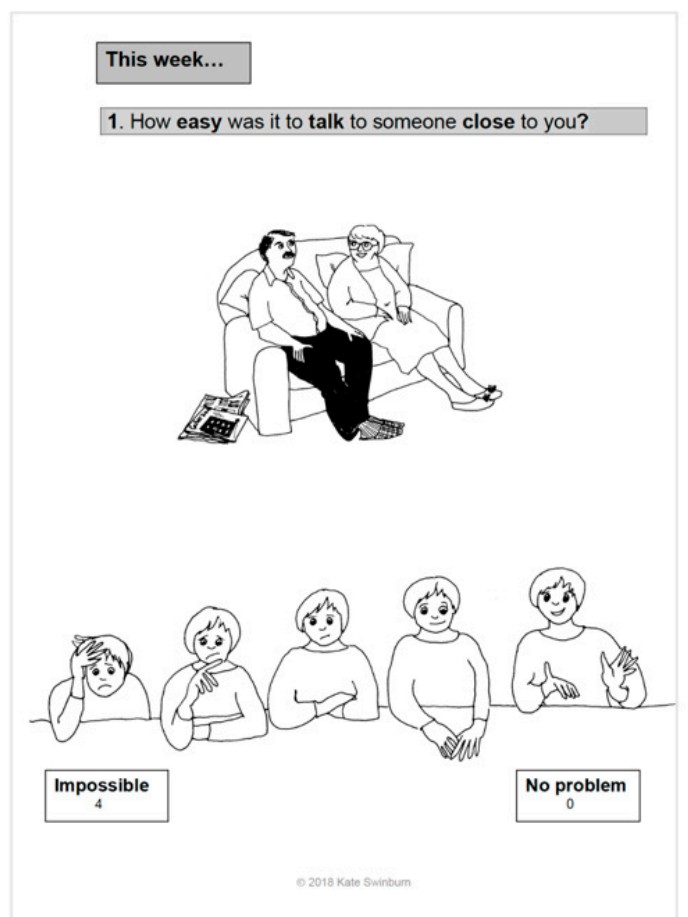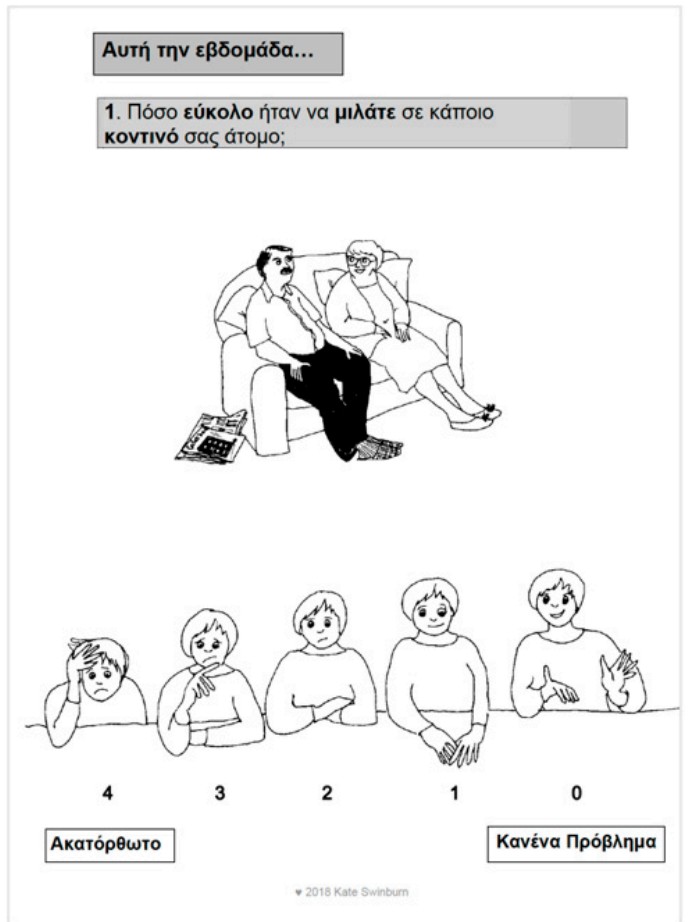

**Figure 1.** Example of item scoring of the British (**left**) and Greek (**right**) version, where the numbers of the Likert Scale were added under each illustration.

To confirm the content validity and cultural relevance of the revised AIQ-21-GR, a second round of cognitive debriefing interviews were conducted [41]. Participants of the first round, were asked to rate the AIQ-21-GR by responding to five statements for each of the 21 items as reported below (see Table 8 in Results Section 3.3.4 for participant responses). The PPI partner A.K. was involved in the development of the statements for the interviews. The statements were as follows: (1) "This item is relevant to my personal situation"; (2) "The item has an appropriate/relevant picture"; (3) "The item's content is clear to me"; (4) "The content of the item is appropriate for me"; and (5) "This is an important item for my life". The range of possible responses were evaluated on a 5-point Likert Scale (1 = strongly disagree; 5 = strongly agree). Cognitive interviews were completed by the authors M.C. and M.S. The statistical analyses of the cognitive interview scores were performed by author P.P., who was blinded to the interview procedure to ensure rigor of the analyses and to prevent bias [40]. See the outcomes in the Results Section 3.3.4, Table 8 on content validity.

The AIQ-21-GR was finalized after a consensus meeting of the main researcher M.C. and PPI partner A.K. where it was agreed that the linguistic content of the items is clear, precise and accessible for the educational level and cultural background of the PWA group.

### 2.9. Pilot Study for Group Discrimination

A pilot study was undertaken to evaluate the acceptability of the finalized version regarding time of administration and whether the tool discriminated between groups. The pilot sample consisted of 23 people with chronic stroke (6+ months post stroke), 15 with aphasia and 8 without. Participants were recruited from the Cyprus Stroke Association

registry and were not included in the validation phase. The AIQ-21-GR was administered to all participants in one session with an average administration time of 20 min. No problems were reported during the administration of the tool. The pilot study was performed to provide an initial indication of whether the AIQ-21-GR can successfully discriminate between groups, as we expected PWA to report greater scores. The pilot results on group discrimination are reported in the Results Section 3.2.

### 2.10. Data Analysis

Reliability and validity of a measure are two properties that indicate the quality and usefulness of a tool. We considered $\alpha = 0.05$ as our threshold for statistical significance [43]. We introduced the common interpretation as proposed by Cohen [43], where 0.2, 0.4, and 0.8 are considered *small*, *medium*, and *strong*, respectively. All statistical analyses of the collected data were analyzed with the jamovi (version 1.6) [44,45] statistics computer software.

### 2.11. Reliability Analyses

In terms of reliability of the AIQ-21-GR, we tested translation reliability and internal consistency of the AIQ-21-GR.

### 2.12. Translation Reliability

Following the multistep translation procedure, a correlation analysis was conducted between the scores of the Greek version and those of the English version as completed by the 16 bilingual SLT final year students. A significant correlation between the two versions was expected to provide evidence for a reliable translation of the AIQ-21 into Greek.

### 2.13. Internal Consistency

Internal consistency indicates the extent to which items of a test measure the same construct (homogeneity). Similar, to the English and Turkish versions, the Cronbach's $\alpha$ coefficient was calculated to test the reliability of the AIQ-21-GR and its items. Cronbach' alpha $\alpha > 0.70$ indicates good internal consistency [46,47]. Like the previous studies measuring the psychometric properties of the AIQ-21, a rounded Cronbach $\alpha \geq 0.8$ was considered excellent [23,27].

### 2.14. Validity Analyses

The following aspects of validity were tested: criterion; construct—with additional factor analysis; content; and known groups validity.

### 2.15. Criterion Validity

Criterion validity refers to how closely an instrument relates to other measures of the same construct [48]. We hypothesized that scores of the AIQ-21-GR was significantly correlated with measures of the QoL, i.e., the SAQOL-39 as there is evidence for links between severity of aphasia with poor QoL [27]. To investigate the validity of the Greek version of the AIQ-21, we tested whether its scores correlated with scores from the SAQOL-39. A negative relationship was expected since negative values indicate greater QoL in AIQ-21 but a lower QoL on the SAQOL-39. To test further the validity of the AIQ-21-GR, we tested correlations between the Communication and Emotional state/Wellbeing domains of the AIQ-21-GR and the Communication and Psychosocial domains of the SAQOL-39 [27]. Moderate correlations were expected since the SAQOL-39 is created for stroke in general, where the AIQ-21 is aphasia specific. Correlational analysis (Spearman's rho) was undertaken to test the validity of the measure. Commonly in psychometric testing [47], correlations between $0 < $ Spearman's $\rho < 0.3$ or $-0.3 < $ Spearman's $\rho < 0$ are considered weak, between $0.4 < $ Spearman's $\rho < 0.6$ or $-0.6 < $ Spearman's $\rho < -0.4$ moderate, and Spearman's $\rho > 0.6$ or Spearman's $\rho < -0.6$ strong.

### 2.16. Construct Validity

Construct validity ensures that the instrument matches and represents the characteristic/construct of the undergoing measurement [7]. The internal validity of the tool was analyzed with correlations between the total and the AIQ-21-GR domains scores. We expect correlations across all domains of the tool and the total score to support the validity of the tool as a measure of QoL in PWA.

### 2.17. Factor Analysis

Similar to previous versions of the AIQ-21 [23,27], we conducted a factor analysis. This analysis provided evidence for the validity of each domain of the AIQ-21: Communication, Participation and Emotional state/Well-being. It was expected that the factors that emerge after a factor analysis should explain around 2/3 of the determined variance [23,27].

### 2.18. Content Validity

Content validity is the degree to which the content of an instrument is an adequate reflection of the construct to be measured [41]. A content validity questionnaire, developed by M.C., M.S. and PPI partner A.K. following the COSMIN guidelines, was scored by a total of 30 participants (4 different groups as described in the Linguistic Adaptation Section 2.8 in Methods). To assess the scores of the COSMIN sub-categories for each item, we analysed the median scores. We expected a median of 4 which shows that participants found the item 'very relevant'. Overall, results are expected to confirm the outcomes of the qualitative cognitive interviews and indicate appropriate content validity of each item of the AIQ-21-GR.

### 2.19. Known-Groups Validity

Known-groups validity determines that an instrument can demonstrate dissimilar scores among different groups [49]. We examined the validity of the AIQ-21-GR in dissociating between PWA and stroke survivors without aphasia. For this analysis, we conducted non-parametric Mann–Whitney *t*-tests comparing overall and domain scores of the AIQ-12-GR of PWA and those without aphasia expecting PWA to score higher than those without aphasia, indicating poorer QoL.

## 3. Results

The reliability and validity of the adapted Greek version of the AIQ-21, was conducted on 69 stroke survivors, of which 47 (mean age = 58.6, sd = 17.5) were PWA with a mean of 19.5 (sd = 20.6) months post stroke diagnosis and 22 (mean age = 61.8, sd = 15.1) were stroke survivors without aphasia with a mean of 22.2 (sd = 19.7) months post stroke diagnosis. Of the 69 participants, 39 (56.5%) were female and 30 (43.5%) were male. The average time post-onset since diagnosis was 20.3 months (sd = 20.5), ranging between 6 and 96 months, indicating that all participants were in the chronic phase post-stroke.

### 3.1. Measure Scores

The descriptive data of the scores for each measure, and their relevant domains, for PWA and stroke survivors without aphasia is presented in Table 3. The median scores of PWA were 3 [$Q_{25} = 2$, $Q_{75} = 4$] for the Aphasia Severity Rating Scale, 37 [$Q_{25} = 22.5$, $Q_{75} = 41$] for the AIQ-21-GR, and 3.51 [$Q_{25} = 3.1$, $Q_{75} = 4.01$] for the SAQOL-39. For stroke survivors without aphasia, the median scores were 13.5 [$Q_{25} = 10$, $Q_{75} = 19.75$] for the AIQ-21-GR, and 3.56 [$Q_{25} = 2.76$, $Q_{75} = 4.1$] for the SAQOL-39.

**Table 3.** Scores in Aphasia Severity Rating Scale (ASRS), AIQ-21-GR, and SAQOL -39 for PWA and stroke survivors without aphasia.

|  | Aphasia | Median | Minimum | Maximum | Percentiles 25th | Percentiles 75th | Whitney-U *t*-Test | *p* Value |
|---|---|---|---|---|---|---|---|---|
| ASRS BDAE-SF | no | - | - | - | - | - | - | - |
|  | yes | 3 | 1 | 5 | 2.00 | 4.00 |  |  |
| AIQ-21-GR total score | no | 13.50 | 0 | 61 | 10.00 | 19.75 | 202 | <0.001 |
|  | yes | 37 | 5 | 67 | 22.50 | 41.00 |  |  |
| AIQ-21-GR Communication | no | 0.00 | 0 | 13 | 0.00 | 5.75 | 143 | <0.001 |
|  | yes | 12 | 1 | 22 | 6.50 | 15.00 |  |  |
| AIQ-21-GR Emotional state/Well-being | no | 8.00 | 0 | 36 | 3.25 | 13.00 | 285 | 0.003 |
|  | yes | 16 | 0 | 36 | 9.00 | 22.00 |  |  |
| AIQ-21-GR Participation | no | 2.50 | 0 | 12 | 0.00 | 6.75 | 283 | 0.002 |
|  | yes | 7 | 0 | 16 | 3.50 | 11.00 |  |  |
| SAQOL-39 total score | no | 3.56 | 1.92 | 4.77 | 2.76 | 4.10 | 517 | 1 |
|  | yes | 3.51 | 1.17 | 4.90 | 3.10 | 4.01 |  |  |
| SAQOL-39 Communication | no | 5 | 2.71 | 5 | 4.18 | 5 | 210 | <0.001 |
|  | yes | 3.86 | 1.29 | 5 | 2.86 | 4.21 |  |  |
| SAQOL-39 Psychosocial | no | 3.84 | 1.88 | 5 | 2.98 | 4.06 | 369 | 0.056 |
|  | yes | 3.5 | 1.63 | 4.88 | 2.88 | 3.72 |  |  |

Note: ASRS BDAE-SF, Aphasia Severity Rating Scale Boston Diagnostic Aphasia Examination- Short Form; Rating: <2 severe aphasia, 3 = moderate aphasia, >4 mild aphasia.

### 3.2. Results Regarding the Reliability of the AIQ-21-GR

3.2.1. Translation Reliability

(a)  Pilot testing for language clarity between Greek and the original English version

Following the linguistic adaptation procedure described in the Method Section, to quantify and test the reliability of the translated version, a correlation analysis was conducted between the scores of the Greek version (median = 13, $[Q_{25} = 8, Q_{75} = 17]$) and those of the English version of the AIQ-21 (median = 11.5, $[Q_{25} = 6.75, Q_{75} = 17]$) by the $n = 16$ (Greek–English bilingual) final year SLT students (see Figure 2). A significant correlation was found (Spearman's ρ = 0.98, $p < 0.001$), which provides robust evidence for a reliable translation of the tool.

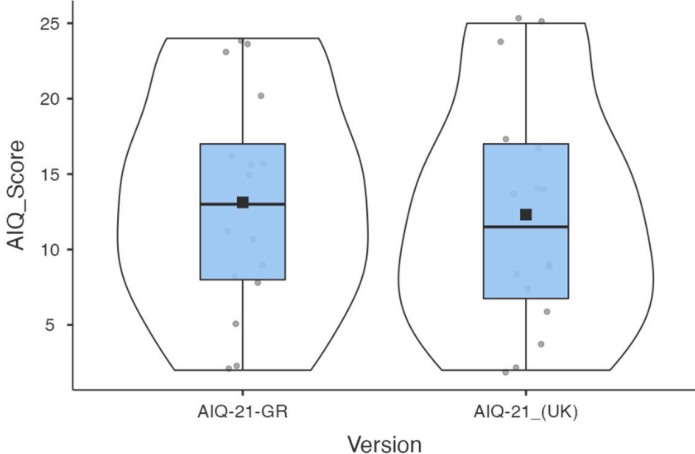

**Figure 2.** Violin Plot of the AIQ-21-GR vs. AIQ-21 (UK) scores for the final year bilingual SLT Students. Note: In Figure 2, the black line indicates the median, while the black square shows the mean for the scores of the final year SLT students in the Greek version of the AIQ-21 (**left**) and the original English version AIQ-21 (**right**). The grey dots indicate individual scores for each version.

(b)　Pilot testing of the final GR version for group discrimination

The finalized version of the AIQ-21-GR was pilot tested for group discrimination with $n = 15$ PWA and $n = 8$ stroke survivors without aphasia. Mann–Whitney U tests indicated that the AIQ-21-GR scores of PWA (median = 52, $[Q_{25} = 41.5, Q_{75} = 64.5]$), compared to those without aphasia (median = 11, $[Q_{25} = 7, Q_{75} = 16.5]$), were significantly greater ($U = 1.50$, $p < 0.001$, $d = 2.93$). This provided an initial indication that the AIQ-21-GR can successfully discriminate between groups, where PWA reported as expected greater scores.

3.2.2. Internal Consistency

The Cronbach's α coefficient was calculated to test the reliability of the AIQ-21-GR and its items (see Table 4). The AIQ-21GR showed an overall excellent reliability (α = 0.914), as well as good reliability for each domain: Communication (α = 0.887), Participation (α = 0.861), and Emotional state/Well-being State (α = 0.892). Further, we tested changes in the overall coefficient after removing each individual item (see Table 4 '*If item is dropped*'), where the reliability of the tool remained excellent (all αs > 0.91). Moreover, we calculated the correlations of each individual item with the total score from the remaining items (see Table 4 '*Item-rest correlation*'), resulting in reliable correlations [50] for all items (all Spearman's ρ = 0.38).

**Table 4.** Reliability analysis of AIQ-21-GR.

| AIQ-21-GR | Cronbach's α | |
|---|---|---|
| Overall | 0.914 | |
| *Domains* | | |
| Communication | 0.887 | |
| Participation | 0.861 | |
| Emotional state/Well-being | 0.892 | |
| *items* | *If item is dropped* | *Item-rest correlation* |
| Item 1 | 0.911 | 0.516 |
| Item 2 | 0.909 | 0.619 |
| Item 3 | 0.911 | 0.513 |
| Item 4 | 0.910 | 0.551 |
| Item 5 | 0.910 | 0.573 |
| Item 6 | 0.909 | 0.594 |
| Item 7 | 0.909 | 0.588 |
| Item 8 | 0.913 | 0.471 |
| Item 9 | 0.909 | 0.611 |
| Item 10 | 0.911 | 0.541 |
| Item 11 | 0.910 | 0.554 |
| Item 12 | 0.911 | 0.507 |
| Item 13 | 0.906 | 0.728 |
| Item 14 | 0.910 | 0.550 |
| Item 15 | 0.912 | 0.475 |
| Item 16 | 0.909 | 0.606 |
| Item 17 | 0.910 | 0.586 |
| Item 18 | 0.909 | 0.622 |
| Item 19 | 0.910 | 0.593 |
| Item 20 | 0.911 | 0.522 |
| Item 21 | 0.914 | 0.378 |

*3.3. Results Regarding the Validity of the AIQ-21-GR*

3.3.1. Criterion Validity

To investigate the validity of the Greek version of the AIQ-21, we tested whether the scores obtained correlated with scores from the SAQOL-39 (see Table 5).

**Table 5.** Spearman rho correlations between AIQ-2-GR and SAQOL-39.

| | | SAQOL-39 | |
| | Total Score | Communication | Psychosocial |
|---|---|---|---|
| **AIQ-21-GR** Total Score | −0.572 *** | | |
| Communication | | −0.624 *** | |
| Emotional state/Well-being | | | −0.516 *** |

Note: *** *p* < 0.001.

As expected, a significant negative correlation (see Figure 3) was found between the scores of the two tools (Spearman's ρ = −0.572, *p* < 0.001). A negative relationship was expected since negative values indicate higher QoL for the AIQ-21-GR but a lower QoL for the SAQOL-39.

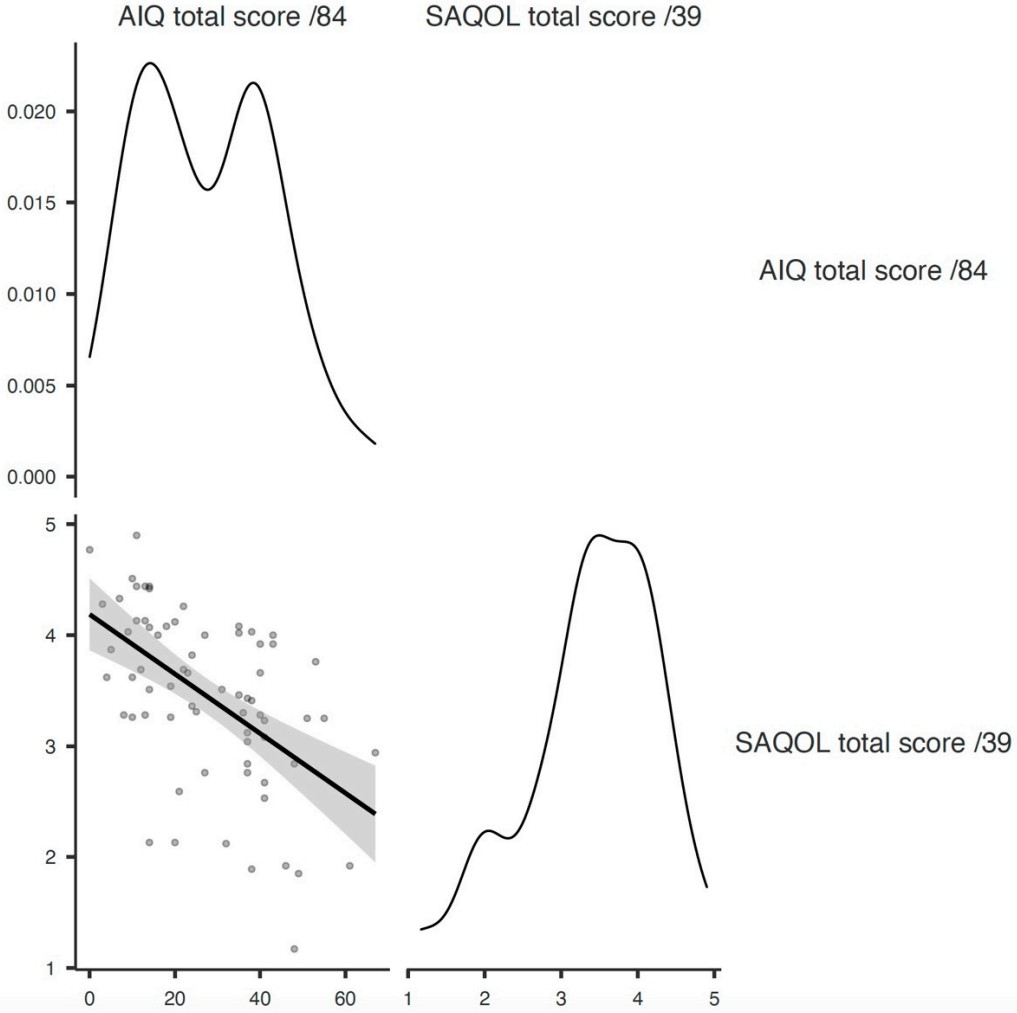

**Figure 3.** Correlation values comparing the total score on the AIQ-2-GR with the total score on the SAQOL-39.

In addition, we further tested correlations between the Communication and Emotional state/Well-being domains of the AIQ-21-GR and the Communication and Psychosocial domains of the SAQOL-39, respectively [27]. Similarly, significant negative correlations were found for the Communication (Spearman's ρ = −0.624, *p* < 0.001), and Emotional state/Well-being and Psychosocial (Spearman's ρ = −0.516, *p* < 0.001) domains of the

two tools. Even though the correlations were moderate and not strong, this significant relationship between AIQ-2-GR and SAQOL-39 serves as evidence that the AIQ can reliably measure the impact of aphasia on QoL.

### 3.3.2. Construct Validity

Furthermore, the construct validity of the tool was analyzed by determining any correlation between the total and the domain scores of the AIQ-21-GR. Overall, significant correlations were found across all domains of the tool (see Table 6 for details), and the total score correlated significantly with the Communication (Spearman's $\rho$ = 0.657, $p < 0.001$), the Participation (Spearman's $\rho$ = 0.601, $p < 0.001$), and the Emotional state/Well-being (Spearman's $\rho$ = 0.732, $p < 0.001$) domains of the AIQ-21-GR. These analyses further support the robustness of the tool to measure QoL in PWA.

**Table 6.** Spearman rho correlations between AIQ-21-GR total score and the three domains of the tool.

| AIQ-21-GR | Communication | Participation | Emotional State/Well-Being |
|---|---|---|---|
| Communication | | 0.463 *** | 0.479 *** |
| Participation | | | 0.460 *** |
| Total Score | 0.657 *** | 0.601 *** | 0.732 *** |

Note: *** $p < 0.001$.

### 3.3.3. Factor Analysis

A factor analysis was conducted using the Maximum Likelihood Method [51]. Three factors were identified corresponding to the three (Communication, Participation and Emotional state/Well-being) domains of the AIQ-21-GR (see Table 7). All but one item was successfully extracted in the corresponding domain. The remaining item, Item 21, was extracted within the Communication domain, instead of the Emotional state/Well-being domain. See the Section 4 for details.

**Table 7.** AIQ-21-GR Factor Analysis using Maximum Likelihood estimates.

| | Factor | | |
|---|---|---|---|
| | Emotional State/ Well-Being | Communication | Participation |
| Item 1 | | 0.923 | |
| Item 2 | | 0.923 | |
| Item 3 | | 0.653 | |
| Item 4 | | 0.552 | |
| Item 5 | | 0.599 | |
| Item 6 | | 0.504 | |
| Item 7 | | | 0.694 |
| Item 8 | | | 0.863 |
| Item 9 | | | 0.812 |
| Item 10 | | | 0.642 |
| Item 11 | 0.698 | | |
| Item 12 | 0.657 | | |
| Item 13 | 0.698 | | |
| Item 14 | 0.546 | | |
| Item 15 | 0.618 | | |
| Item 16 | 0.836 | | |
| Item 17 | 0.752 | | |
| Item 18 | 0.710 | | |
| Item 19 | 0.688 | | |
| Item 20 | 0.434 | | |
| Item 21 | | 0.307 | |

### 3.3.4. Content Validity

Based on the COSMIN guidelines for the evaluation of relevance, comprehensiveness and comprehensibility of all items, a questionnaire on content validity was scored by $n = 30$ participants, from the translation validation group, as described in the Method Section. Participants included $n = 10$ healthy individuals, $n = 6$ PWA, $n = 10$ SLTs and $n = 4$ stroke survivors without aphasia. The median scores for each group on the subcategories of the COSMIN are presented in Table 8.

**Table 8.** Median Scores of Each Group in Each Subcategory of the COSMIN.

|  | | | Percentiles | |
|---|---|---|---|---|
|  | Group_Adj | Median | 25th | 75th |
| Relevance | Healthy | 4.00 | 3.25 | 4.75 |
|  | PWA | 5.00 | 5.00 | 5.00 |
|  | SLTs | 4.00 | 4.00 | 5.00 |
|  | Stroke no aphasia | 4.00 | 3.75 | 4.25 |
| Appropriateness | Healthy | 4.00 | 3.25 | 4.00 |
|  | PWA | 5.00 | 5.00 | 5.00 |
|  | SLTs | 4.50 | 4.00 | 5.00 |
|  | Stroke no aphasia | 4.00 | 3.75 | 4.25 |
| Importance | Healthy | 5.00 | 4.25 | 5.00 |
|  | PWA | 5.00 | 5.00 | 5.00 |
|  | SLTs | 5.00 | 5.00 | 5.00 |
|  | Stroke no aphasia | 4.00 | 4.00 | 4.25 |
| Clarity | Healthy | 4.00 | 4.00 | 4.75 |
|  | PWA | 5.00 | 5.00 | 5.00 |
|  | SLTs | 5.00 | 4.00 | 5.00 |
|  | Stroke no aphasia | 4.00 | 4.00 | 4.25 |
| Content | Healthy | 4.00 | 4.00 | 4.75 |
|  | PWA | 4.50 | 4.00 | 5.00 |
|  | SLT | 4.50 | 4.00 | 5.00 |
|  | Stroke no aphasia | 4.00 | 4.00 | 4.25 |

In total, content validity was given higher scores with an overall median score of 4 [$Q_{25} = 4$, $Q_{75} = 5$]. No between group differences were found for the scores of the COSMIN subcategory, namely relevance to personal situation ($\chi^2 = 6.63$, $p = 0.357$), appropriateness/relevance of picture ($\chi^2 = 11$, $p = 0.088$), importance of Item ($\chi^2 = 4.84$, $p = 0.184$), clarity of wording ($\chi^2 = 8.44$, $p = 0.208$), and content appropriateness ($\chi^2 = 7.31$, $p = 0.605$). Overall, these results confirm the outcomes of the qualitative cognitive interviews and indicate appropriate content validity (relevance, comprehensiveness and comprehensibility) for each item of the AIQ-21-GR.

### 3.3.5. Known-Groups Validity

Finally, we examined the validity of the AIQ-21-GR in dissociating between PWA and stroke survivors without aphasia. Specifically, we conducted Mann–Whitney U $t$-tests comparing overall and domain scores of the AIQ-21-GR of PWA and those without aphasia. For the overall scores, PWA (median = 37, [$Q_{25} = 22.5$, $Q_{75} = 41$]) compared to stroke survivors without aphasia (median = 13.5, [$Q_{25} = 10$, $Q_{75} = 19.8$]), scored significantly higher ($U = 202$, $p < 001$, $d = 1.143$), indicating poorer QoL, as expected (see Figure 4A). Similar results were found between the scores of PWA (median = 12, [$Q_{25} = 6.5$, $Q_{75} = 15$]) and people without aphasia (median = 0, [$Q_{25} = 0$, $Q_{75} = 5.75$]) for the Communication domain ($U = 143$, $p < 001$, $d = 1.419$), for PWA (median = 7, [$Q_{25} = 3.5$, $Q_{75} = 11$]) and people without aphasia (median = 2.5, [$Q_{25} = 0$, $Q_{75} = 6.75$]) for the Participation domain ($U = 283$, $p = 0.002$, $d = 0.810$), and for PWA (median = 16, [$Q_{25} = 9$, $Q_{75} = 22$]) and people without aphasia (median = 8, [$Q_{25} = 3.25$, $Q_{75} = 13$]) for the Emotional state/Well-being domain ($U = 285$, $p = 0.003$, $d = 0.696$) of AIQ-21-GR (see Figure 4B). In Figure 4A,B, the black line

indicates the median, while the black square shows the mean for stroke survivors without aphasia (left) and those with aphasia (right). The grey dots indicate the individual scores for each participant.

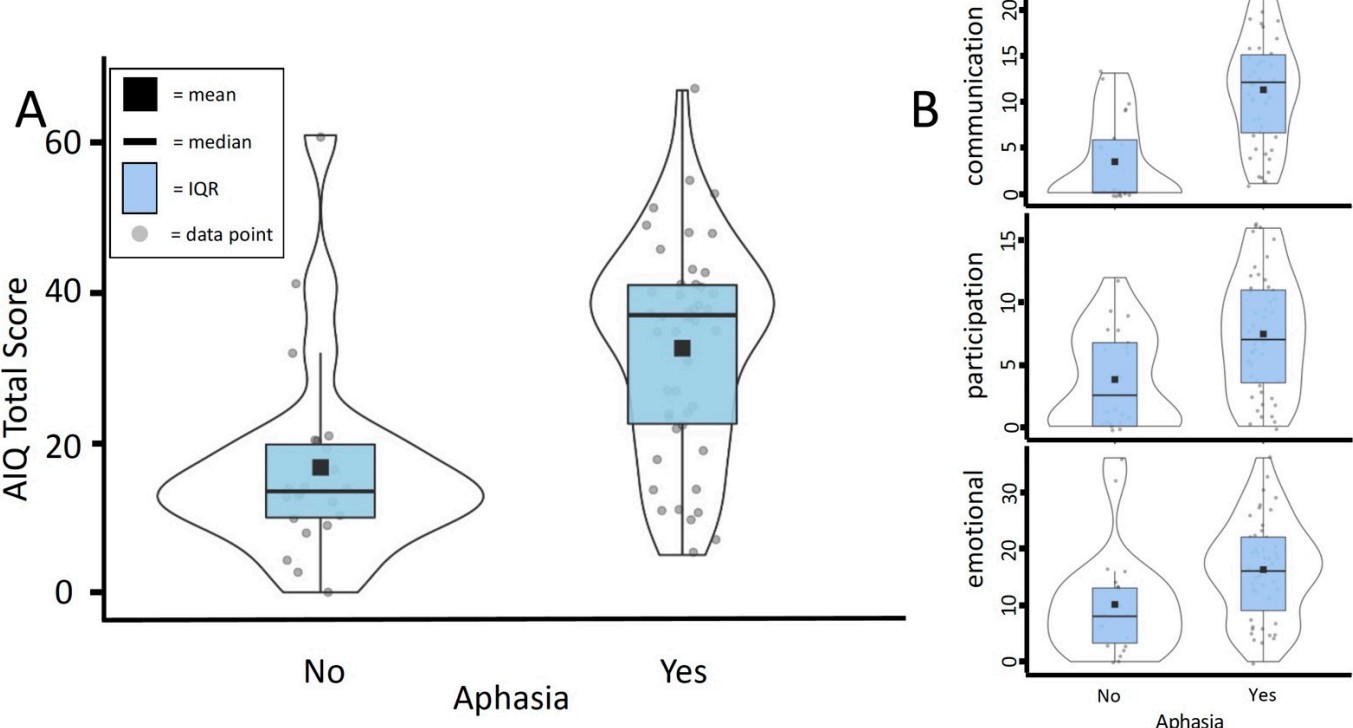

**Figure 4.** Violin Plot of the Scores of PWA compared to stroke survivors without aphasia on the AIQ-21-GR. (**A**) shows the overall scores of PWA compared to stroke survivors without aphasia. (**B**) shows the results between the scores of PWA and people without aphasia on the three domains of the AIQ-21-GR.

## 4. Discussion

In this study, we translated and adapted the AIQ-21 into Greek (AIQ-21-GR). Participants were prompt to self-rate an aphasia-friendly questionnaire that evaluated the impact of aphasia on the QoL of chronic PWA. The quantitative results of this study fully support the reliability and validity of the AIQ-21-GR adaptation.

### 4.1. Regarding the Reliability of the AIQ-21-GR

The results of the reliability testing of the AIQ-21-GR provide evidence for a reliable translation of the tool into Greek. The significant correlation found between the reliability of the Greek translation versus the original English version confirms the above statement. This outcome is in agreement with the results in the study of Yaşar et al. [27] for the adaptation and validation of the AIQ-21 into Turkish. This study also confirms that the AIQ-21-GR has significantly high internal consistency which is again consistent with the findings from the original UK study [23] and the recent Turkish [27] version of the tool.

### 4.2. Regarding the Validity of the AIQ-21-GR

The results of our study support that the Greek version of the AIQ-21 has excellent validity. Criterion validity testing revealed a significant negative correlation between the scores of the AIQ-21-GR with scores of the Greek version of the SAQOL-39, a result which comes in agreement with the scores obtained during the Turkish adaptation study of the AIQ-21-TR [27]. Further significant negative correlations between the Communication and Emotional state/Well-being domains of the AIQ-21-GR with the equivalent Communication

and Psychosocial domains of the SAQOL-39 were also confirmed as previously [27]. Even though the correlations were moderate and not strong, this significant relationship between AIQ-12-GR and SAQOL-39 serves as evidence of the construct validity of the AIQ-21-GR to measure QoL.

Moreover, construct validity testing via a factor analysis on the 21 items of the AIQ-21-GR has shown that the three factors were identified corresponding to the three domains of the AIQ-21-GR: Communication, Participation and Emotional state/Well-being. For this study, all but one item (95%) were successfully extracted in the corresponding domain. Item 21 'How do you feel about the future', with responses ranging from 4 = very negative to 0 = very positive, was extracted within the Communication domain, instead of the Emotional state/Well-being domain. One interpretation is that the vagueness of the question, which can have multiple interpretations, probably lead to the lower factor estimates. This might be also because the '*future*' is an abstract word, and according to Kiran et al. [52], abstract words have low imageability and concreteness and PWA exhibit an exaggerated concreteness effect. Borghi [53] stated that abstract words are grounded not only by our sensorimotor experiences, such as concrete concepts, but also by the linguistic, social, and inner experiences of each person. This might also be linked to the fact that PWA who experience constant challenges with functional communication show abstract word deficits in conveying and understanding complex ideas, i.e., anticipating for their '*future*' [54]. Therefore, from the results of the factor analysis, PWA might associate their '*future*' with their personal experiences while living with aphasia and not in a general sense, and this is the reason the item moved from the Emotional state/Well-being domain to the Communication domain. This finding shows how PWA might reflect on their future based on the Activity and Participation [17] domains, highlighting barriers they face during ADL execution due to their aphasia [6]. Overall, the factor analysis is very good and close to the original version by Swinburn et al. [23].

Furthermore, the known groups validity of the AIQ-21-GR in discriminating between PWA and stroke survivors without aphasia was also true for this study. Specifically, when we compared the overall and domain scores of the AIQ-21-GR of PWA to stroke survivors without aphasia, scores were significantly higher to PWA indicating poorer QoL as expected. These results revealed that stroke survivors without aphasia rate themselves as having a higher QoL compared to the PWA group. Again, this result comes in agreement with previous studies that confirm that aphasia impacts the QoL of PWA negatively compared to stroke survivors without aphasia [55].

Finally, the results from the cognitive interviews based on the COSMIN guidelines [40] indicated that total content validity of the AIW-21-GR was given high scores. This confirms the positive outcomes of the cognitive interviews and indicates the appropriate content validity of each item concerning their relevance, comprehensiveness, and comprehensibility to PWA.

### 4.3. Patient and Public Involvement

In this study, we emphasized the active involvement of PWA and followed co-production methodology throughout the translation and adaptation phase [23,56]. PPI methodology was implemented to ensure the tool's relevance, appropriateness, and test its content validity [3]. The involvement of co-author A.K., the PPI partner, in the research procedure optimized the validity and applicability of the research itself and the effectiveness of the resulting tool [18]. A.K. was actively involved in evaluating and overseeing the development of the translated versions and was greatly engaged during the validation of its content and the finalization of the questionnaire. Reflecting on the active engagement of A.K. as a PPI partner in this study, we report that her involvement from the beginning of the study had a positive influence as she helped the team create a questionnaire with real-life content to PWA. From her end, A.K. reported a positive experience of her involvement in the study as researcher and felt well-supported by the team. Nevertheless, A.K. reported that during the consensus meeting for the finalization of the Greek version of the AIQ-21, she often experienced fatigue because of the large volume of verbal and written material

she had to process. This was despite the plenty of time given to A.K. to review the materials, identify changes, record her comments, reflect on the content, and respond at her own pace, a process that made the meetings timely and very laborious.

### 4.4. Clinical Implication

The AIQ-21-GR is a psychometrically sound patient reported outcome measure proven to be responsive to the areas PWA identify as key to QoL after stroke. It is recommended that the AIQ-21-GR can be used in clinical practice for information gathering, functional goal setting and as an outcome measure of QoL. The AIQ-21-GR is a short (consists of 21 items) aphasia-friendly PROM tool that can be administered in approximately 25 min. It can be utilized from the rehabilitation setting to community reintegration centres with people with chronic communication challenges after stroke. This tool informs the practicing clinician and/or researcher about the impact of the language impairment on QoL from the perspective of the person with aphasia. The AIQ-21-GR demonstrates subjective lived experiences around the Activity and Participation domains of the ICF [17]. According to Swinburn et al. [23] 'The AIQ-21 is a PROM that has great potential to be one of the core set of aphasia tests for clinical and research use' (p.23). The outcomes of the AIQ-21-GR can be used alongside impairment-based language assessments used by clinicians in the acute and subacute phase of rehabilitation to enable functional goal setting in close collaboration with people with aphasia. The AIQ-21-GR as a validated PROM in Greek aims to empower people living with chronic aphasia by revealing and acknowledging the communication challenges and the impact of aphasia on each person's everyday life [23].

### 4.5. Limitations of the Study

A limitation is that the participants were experiencing life with chronic aphasia across a broad range of years post stroke (6 months to 8 years post-stroke). This wide aphasia chronicity timeline may have affected how PWA rated QoL. According to recent studies, people enter the chronic phase (6 months post stroke) from a rehabilitation setting, might not have been able to adjust and self-manage their communication difficulties and psychosocial challenges after the stroke [57]. Nichol and colleagues [57] described such self-management behaviours as 'seeking information, managing symptoms, addressing psychological issues, and supporting lifestyle/social changes and communication' (p.3). Since most of these topics are covered in the three domains of the AIQ-21, participants might have rated themselves lower than those who have achieved self-management and have developed compensatory strategies for their communication impairments at later stages, e.g., >2 years post stroke.

### 4.6. Future Directions

One recommendation is that future research on the tool involve people who live with various types and aetiologies of aphasias and not only on stroke-based aphasias. Stroke is characterised by the stability of the neurological symptoms and PWA often live in the chronic phase with little or no further deterioration of their language and communication skills. Therefore, people with dementia, primary progressive aphasia or other neurodegenerative diseases that result in aphasia, could use the AIQ-21-GR to self-evaluate their QoL regularly and stimulate functional goal setting and individualized interventions in close collaboration with their clinicians.

## 5. Conclusions

The AIQ-21-GR is a reliable and valid tool that assesses QoL in people with chronic aphasia. The psychometric properties of the AIQ-21-GR are consistent with the psychometric qualities of the original AIQ-21 UK study [23] and the Turkish study for the adaptation of the AIQ-21-TR [27]. The AIQ-21-GR is a very useful clinical tool and a reliable patient reported outcome measure for examining aphasia impact-related QoL, which can guide patient-centered interventions, and promotes functional goal setting.

**Author Contributions:** Conceptualization, M.C. and M.K.; methodology, M.C., J.-M.A. and M.K; software, P.P.; validation, M.C., A.K., P.P. and M.S.; formal analysis, P.P., M.C., A.K. and M.S; investigation, M.C., M.S. and L.P.; resources, M.C., A.K. and M.K; data curation, M.C., P.P. and M.K.; writing—original draft preparation, M.C.; writing—review and editing, M.C., J.-M.A. and M.K.; J.-M.A. and M.K; project administration, M.C. and M.S.; funding acquisition, M.C. All authors have read and agreed to the published version of the manuscript.

**Funding:** This work was supported by the A.G. Leventis Foundation Doctoral Full Scholarship Grant, Geneva, Switzerland. Funding Number: 19298.

**Institutional Review Board Statement:** For the validation of the AIQ-21-GR ethical approval was obtained from the Cyprus National Bioethics Committee (EEBK/ΕΠ/2017/37).

**Informed Consent Statement:** Informed consent was obtained from all subjects involved in the study.

**Data Availability Statement:** The data generated during or analyzed in the current study are not publicly available due to ethical restrictions. All data queries and requests should be submitted to the corresponding author, Marina Charalambous PhD Researcher, for consideration.

**Acknowledgments:** The authors thank all the people with stroke and aphasia, and their families, for their excellent collaboration and support to this research. We also thank Androulla Athanasiou from the Language Center of the Cyprus University of Technology, Ioanna Orphanidou, linguist and clinician in Athens Greece and PhD researcher at the Department of Rehabilitation Sciences Cyprus University of Technology, and Elena Theodorou from the Department of Rehabilitation Sciences Cyprus University of Technology for their input and assistance with the Greek translations.

**Conflicts of Interest:** The authors declared no potential conflicts of interest with respect to the research, authorship, and/or publication of this article.

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
