# Peer review of "Adaptation of the Aphasia Impact Questionnaire-21 into Greek: A Reliability and Validity Study"

_ctn, doi:10.3390/ctn6040024_

Round 1
Reviewer 1 Report
This study investigated to determine the reliability and validity of the Greek 21 version of the AIQ-21. As a result, high reliability and validity were reported. This study was significant for Greek aphasic patients. However, I have some concerns.
3 of 5, line91
AIQ-21-GR was translated using PPI methodology. I think this methodology needs to be added to this section.
3 of 5, line 115
The sample size was compared to previous study. Why don't you calculate the sample size?
Table
Table's frame is large. Please make it more compact and easier to read.
Results
9 of 5, line 317-328
Are their significantly difference in 2 group ?
Reviewer 2 Report
This manuscript details the translation of the Aphasia Impact Questionnaire-21 (AIQ-21), a patient-reported outcome measure for aphasia, into Greek. Validation for the AIQ-21 in Greek was compared to the gold-standard tool for measuring quality of life in Greek, called the Stroke and Aphasia Quality of Life Scale-39 (SAQOL-39). Overall, the manuscript describes methods that are well motivated, well carried out, and aptly described. All comments below are minor.
Line 102: Please use the previously defined acronym “SLT” instead of “speech and language therapist,” if applicable.
Data Analysis: For each section, please ensure that the following details are specified regarding statistical significance evaluations: Correlation type (e.g., Pearson’s vs. Spearman’s) or other statistical testing type, alpha level for statistical significance, cut-offs for correlation strength, measures for effect size (please provide where applicable, I see Cohen’s d is provided for Mann-Whitney U tests later on (section 3.2.5) but without much interpretation). Some of these details are provided in one section but not others (e.g., Criterion Validity uses Spearman’s rho with clear cut-offs defined, but Translation Reliability is missing this information).
Figure 3 is quite blurry—is it possible to include a higher-resolution version?
The axes in Figure 4B are too small and very blurry—is it possible to include these subplots as a separate figure or increase the resolution of 4B as it stands?
